# Quantifying mantle mixing through configurational Entropy

Erik van der Wiel[1], Cedric Thieulot[1], Douwe J.J. van Hinsbergen[1]

[1] Department of Earth Sciences, Utrecht University, Princetonlaan 8A, 3584 CB Utrecht, the

Netherlands

Corresponding author: e.vanderwiel@uu.nl

# 1 Abstract

Geodynamic models of mantle convection provide a powerful tool to obtain insights
into the structure and composition of the Earth's mantle that resulted from a long history of
differentiating and mixing. Comparing such models with geophysical and geochemical
observations is challenging as these datasets often sample entirely different temporal and
spatial scales. Here, we explore the use of configurational entropy, based on tracer and
compositional distribution on a global and local scale. We show means to calculate
configurational entropy in a 2D annulus and find that these calculations may be used to
quantitatively compare long-term geodynamic models with each other. The entropy may be
used to analyze, with a single measure, the mixed state of the mantle as a whole and may
also be useful to compare numerical models with local anomalies in the mantle that may be
inferred from seismological or geochemical observations.

## 14 1. Introduction

Mantle convection models that are used to simulate the evolution and dynamics of
the solid Earth are built on different sets of observations each providing their own
constraints to validate the state of the mantle through time (e.g., (Dannberg & Gassmöller,
2018; Gerya, 2014). For instance, with the advent of full-plate kinematic reconstructions of
the past 100s Ma (e.g., (Domeier & Torsvik, 2014; Merdith et al., 2021), mantle models can
now be driven by plate motions through geological time (e.g., (Coltice & Shephard, 2018;
Flament et al., 2022; Heister et al., 2017). Such experiments then lead to a prediction of the
structure and composition of the mantle that may be compared to geological, geochemical,
or seismological observations from the modern Earth (e.g. (Bower et al., 2015; Flament et
al., 2022; Li et al., 2023; Lin et al., 2022; Yan et al., 2020).

Key observables of the modern Earth that may be predicted by models are anomalies

in mantle structure or composition that result from mantle mixing, or absence thereof. For
instance, seismic tomography provides images of the present-day mantle as relative slow
and fast regions in terms of seismic wave propagation, which can relate to variations in
temperature and/or composition such as slabs or mantle plumes (Koelemeijer et al., 2017;
Ritsema & Lekić, 2020). The heterogeneity of the Earth's mantle is also reflected in
geochemical observations of magmatic rocks, oceanic island basalts (OIB) and mid-oceanic
ridge basalt (MORB), which suggest the existence of depleted, enriched, and even primordial
mantle reservoirs, i.e. unmixed regions that maintain a geochemically distinct composition
(Jackson et al., 2018; Jackson & Macdonald, 2022; McNamara, 2019; Stracke et al., 2019).
Notably, seismological and geochemical heterogeneities may entirely, partly, or hardly
overlap, and observations may relate to spatial and temporal scales. Seismology reveals
seismic velocity anomalies in the mantle on scales of 100s to 1000s of km, varying from slabs
to LLSVPs (e.g., (Garnero et al., 2016; Ritsema et al., 2011; van der Meer et al., 2018).
Geochemical differences between MORBs from the Atlantic, Pacific and Indian Ocean
indicate compositional heterogeneity on a hemispheric scale (Doucet et al., 2020; Dupré &
Allègre, 1983; Hart, 1984; Jackson & Macdonald, 2022), geochemical zonation within a single
plume system is evidence for heterogeneities on a 100 km scale (Gazel et al., 2018; Hoernle
et al., 2000; Homrighausen et al., 2023; Weis et al., 2011), whereas micro-scale analysis
reveals even major variations between samples (Stracke et al., 2019). All such variations may
result from a cycle of geochemical differentiation and renewed mixing that is associated with
mantle convection and that eventually may be predicted by mantle convection models. To
this end it is important to also able to define or quantify the mixed state of the modern
mantle from a suitable numerical mantle convection model on the relevant range of spatial
scales. While mixing technically involves diffusion at small scales and the term stirring has
been proposed to account for the mechanical stretching and folding (Farnetani & Samuel,
2003), which is infact our interest here, we shall nevertheless use the term mixing in the
remainder of the manuscript as we use varying 'compositions' that are able to mix.

It has long been recognized that mantle convection is complex, and its mixing has

been studied for decades, see (Kellogg, 1993; van Keken et al., 2003) for early reviews on
this topic. Unsurprisingly, the advent of high-performance numerical modelling in the mid-
90's saw a resurgence in the characterization of mantle mixing and its quantification. Various
approaches have been proposed over the years, but the vast majority of these are based on
the time evolution of a swarm of particles. Early studies (such as (Hoffman & McKenzie,
1985; Olson et al., 1984a, 1984b; Richter et al., 1982; Schmalzl et al., 1996)) use statistics to
arrive at a mixing time scale. Another approach using the presence, addition, and/or
removal of particles in a modelled domain is used to quantify mixing-times and degassing
(sampling of primitive mantle) (Gottschaldt et al., 2006; Gurnis & Davies, 1986a, 1986b), to
measure strain and the dispersal of tracers (Christensen, 1989; Kellogg & Turcotte, 1990) or
to study the development of time-dependent mantle-heterogeneities (Hunt & Kellogg,
2001). Note that other methods have been proposed, such as a line method (Ten et al.,
1998), a correlation dimension method (Stegman et al., 2002) and a hyperbolic persistence
time method (Farnetani & Samuel, 2003).

More recently another approach has dominated the mantle mixing literature: it

consists in measuring the Lyapunov time, which is the characteristic timescale for which a
dynamical system is chaotic, or rather its inverse the Lyapunov exponent. It can be shown
that mixing is laminar or turbulent by evaluating the Lyapunov exponent, the larger the
exponent the more efficient the mixing is. A typical example uses a steady state velocity
pattern obtained in a 3D spherical domain to advect passive particles (van Keken & Zhong,
1999). They use a very common approximation to the Lyapunov exponent, i.e., the Finite
Time Lyapunov Exponent, which is based on the evaluation of the distance between a
multitude of particle pairs that are initially very close to each other (i.e., stretching of this
original distance after 4 Ga). This shows a strong diversity in mixing behavior dependent on
the mantle flow characteristics. Other studies that used the same approach in studying a
variety of mantle convection problems include (Bello et al., 2014; Bocher et al., 2016; Colli et
al., 2015; Coltice, 2005; Coltice & Schmalzl, 2006; Farnetani et al., 2002; Farnetani & Samuel,
2003; Ferrachat & Ricard, 1998, 2001; Samuel et al., 2011; Tackley & Xie, 2002; Thomas et
al., 2024).

In this study, we investigate the merits of yet another approach, the configurational

(or 'Shannon') entropy for quantifying compositional mixing of particles through flow on a
global or local scale (Shannon, 1948). Despite its popularity in other fields it has only
minimally been used in geosciences (e.g., (Camesasca et al., 2006; Naliboff & Kellogg, 2007).
We develop the application of configurational entropy to the 2D cylindrical mantle
convection models which we recently developed (van der Wiel et al., 2024), implementing
measures for local and global entropy of mixing that incorporates information on
composition. We aim to use configurational entropy to quantify the degree of mixing on
different scales for different hypothetical initial compositional configurations of the mantle
and evolution thereof over time. Subsequently, we discuss how configurational entropy may
be used as a bridge for quantitative comparison between mantle convection models and
geological, seismological, or geochemical observations.

## 2.    Methods

### 2.1. Mixing entropy

Configurational entropy is analogous to the Shannon entropy (Shannon, 1948) and related to the probabilities derived from the distribution of particles with a certain value, i.e., composition. It can be used to track the mixing of particles independently of the physical process causing that mixing in numerical simulations as well as laboratory experiments. This entropy is widely used and has a large variety of applications, including fluid or magma mixing (Camesasca et al., 2006; Naliboff & Kellogg, 2007; Perugini et al., 2015), transport of plastic in oceans (Wichmann et al., 2019), distribution of seismicity in earthquake populations (Goltz & Böse, 2002), or the quantification of uncertainty in geological models (Wellmann & Regenauer-Lieb, 2012).

The definition of the configurational entropy $S$ is based on the proportion of a specific distribution of particles in a domain tessellated by non-overlapping cells. For this we use passive particles, or tracers, that are advected in a flow model leading to particle trajectories. The entropy depends on the distribution of particles, the number of cells and the initial compositional distribution (see section 2.3). Let C be the number of compositions and M the number of cells in the domain. The entropy is calculated based on the discretized particle density $\rho_{c,j}$ (Eq. 1), i.e., the amount of particles of composition $c$ in cell $j$,

$$\rho_{c,j} = \frac{n_{c,j}}{N_c} \tag{1}$$

where $N_c$ is the total number of c-particles divided by the number of cells M. This assumes that cells are of equal area, which will be used here in our 2D application. Hence, $N_c$ is the same for all cells. From the compositional density $\rho_{c,j}$ we calculate $P_{j,c}$ which is the proportion of particles of composition c in cell j relative to the total number of particles in

the cell, both measured in terms of density through Eq. (2). We calculate $P_j$ through the cell-
sum of all compositional densities in Eq. (3). $P_j$ is the proportion of the amount of particles
in a cell relative to all the particles in the system. The quantities we describe here as
proportions would be considered probabilities, or conditional probabilities, in statistical
physics.
$$P_{j,c} = \frac{\rho_{c,j}}{\sum_{c=1}^{C} \rho_{c,j}} \qquad\qquad (2)$$
$$P_j = \frac{\sum_{c=1}^{C} \rho_{c,j}}{\sum_{j=1}^{M} \sum_{c=1}^{C} \rho_{c,j}} \qquad\qquad (3)$$
Next, Eq. 4 defines the global entropy $S_{pd}$ of the particle distribution.
$$S_{pd} = -\sum_{j=1}^{M} P_j \ln P_j \qquad\qquad (4)$$
which quantifies the global spatial heterogeneity of the particle distribution independent of
composition (Naliboff & Kellogg, 2007). At the cell level, the local entropy $S_j$ for cell $j$ can be
defined for the mixture of particles with different compositions:
$$S_j = -\sum_{c=1}^{C} P_{j,c} \ln P_{j,c} \qquad\qquad (5)$$
Finally, the global entropy $S$ of the particle distribution, accounting for composition, is the
weighted average of $P_j$ (Eq. 3) and the local entropy $S_j$ (Eq. 4) through Eq. (6) (Camesasca et
al., 2006).
$$S = \sum_{j=1}^{M} P_j S_j \qquad\qquad (6)$$
Maximum entropy is achieved when all particle densities $\rho_{c,j}$ are equal, i.e., the
distribution of composition and number of particles are the same in each cell. Each entropy
above has a different maximum which depends on either the number of cells for ($S_{pd}$) or the
number of compositions used (for $S$ and $S_j$). To compare entropies between mixing models
with different initial conditions, we normalize the entropies by dividing each by its
maximum. The maximum for $S_{pd}$ is equal to $\ln M$, while for $S_j$ and $S$ the maximum is $\ln C$
(Camesasca et al., 2006). This provides values for all entropies between the endmembers 0
(entirely segregated composition) and 1 (uniformly mixed). The maximum value for $S_{pd}$ can
only be reached when all compositions are present in equal ratios. Entropy calculations of
four simple educational examples are shown and explained in appendix A to help the reader
appraise these quantities.

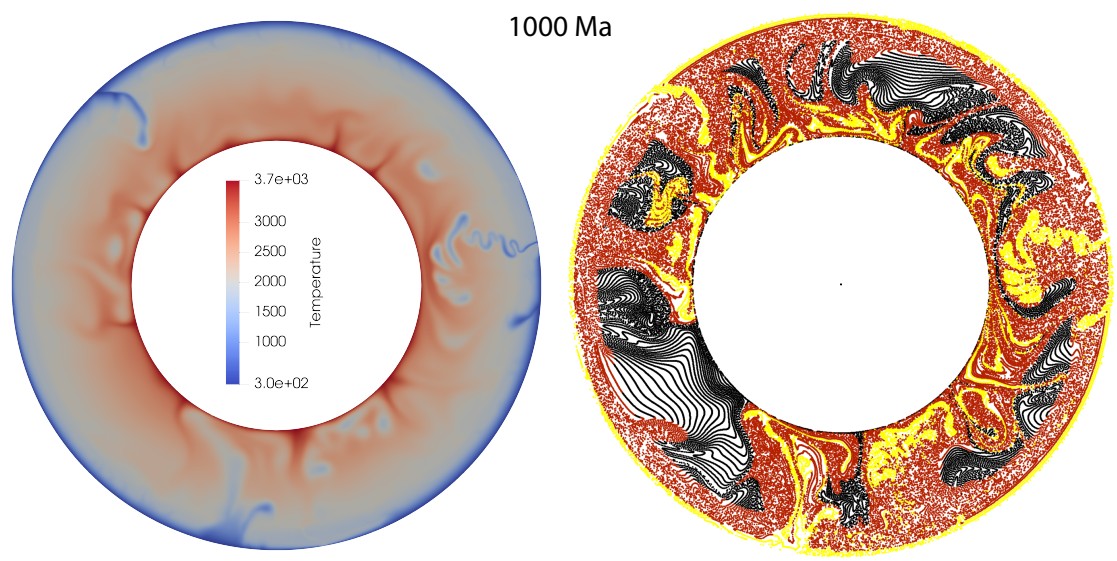


*Figure 1 - Snapshot from model R at t=1000 Ma showing the temperature field in Kelvin (left) and passive particles colored*
*by composition (lithosphere: yellow, upper mantle: red, lower mantle: black).*
**2.2 Mantle convection model**
We apply the configurational entropy to the quantification of mixing in a recently
developed 2D numerical mantle model in a 2D cylindrical geometry that simulates 1000 Ma
of ongoing mantle convection and subduction (van der Wiel et al., 2024). The convection
model was designed to evaluate the sensitivity of inferred lower mantle slab sinking rates
(van der Meer et al., 2018) to the vigor of mantle convection. The simulations comprised
dynamically self-consistent one-sided subduction below freely moving, initially imposed,
continents at the surface, culminating in slab detachment followed by sinking of slab
remnants across the lower mantle (Fig. 1). The surface velocity in the model were generally
between 1 and 4 cm/a, which may be compared to the reconstructed values of 4 cm/a of
Zahirovic et al., (2015) and the obtained average slab sinking rates were in the range of
those that were inferred from correlations between the location of imaged lower mantle
slabs and their geological age (van der Meer et al., 2018). This modelling qualitatively
illustrated the degree of mixing in a modelled mantle and the potential preservation of an
unmixed original mantle, advected slabs, or (partly) homogenized, mixed mantle shown by
the distribution of particles.

We quantify the degree of mantle mixing in the model by investigating the local and

global mixing entropies (Section 2.1) for model *R* of van der Wiel et al. (2023) at different
resolutions. We also illustrate how mixing entropy quantifies the mixing of a different model
(model P) that showed significantly higher slab sinking rates than inferred for the lower
mantle and that displayed a higher degree of mantle mixing (van der Wiel et al., 2023). For
this purpose, we only used the passive particle distribution available from the models in van
der Wiel et al., 2023. The cells used to calculate the configurational entropy (see section 2.4)
are independently substantiated and therefore not the same as used in the numerical
model, for any additional information of these models we refer the reader to van der Wiel et
al., 2023.

**2.3 Initial composition**

To illustrate how we track compositional evolution with configurational entropy, we

assign a compositional distribution to our example models with two different approaches. In
case A, we assign a compositional distribution in the initial model, and each tracer will keep
its initial composition through time. We divide the annulus in two concentric parts at a
radius of 5100 km and assign the inner and outer part a different composition (simply put: a
different color). This creates a 50-50 ratio between the number of particles of each
composition. Case B uses dynamic compositions, i.e., the composition of a particle may
change over time. We use three compositions whose relative ratios are allowed to change
over time depending on the particle's depth in the model. Initially, we define particles as
lower mantle when they start below 660 km depth in the model, upper mantle if they start
between 100 and 660 km, and lithosphere if between 0 and 100 km depth. Particles keep
their 'lower mantle' composition as long as they do not ascend above the 660 km during
model evolution. Any particle that moves from a deeper reservoir into a shallower one will
see its composition changed to the shallower reservoir and will maintain this composition
for the remainder of model time. This approach is an example that may be used in a study to
characterize the secular geochemical differentiation of the solid Earth.

**2.4 Cell distribution**

Entropy as calculated in this study also depends on the number and distribution of

cells, which is independent of the mesh used in the numerical model itself. To ensure an
approximately equal cell-area throughout our domain, we vary the number of cells per radial
layer. The cell-area is determined by the product of the radial extent $\delta r$ and lateral extent $\delta \theta$
that follows from the number of radial layers and the number of cells along the core-mantle-
boundary (CMB) circumference. Varying cell-area may be important to compare the
outcome of a numerical model with datasets that have very different resolutions (e.g.,
seismology versus geochemistry). We illustrate different cell resolutions with our 2D
example model, but a similar approach may be used for a 3D model albeit with a different
tessellation (Thieulot, 2018). One should note that in this set-up our cells are chosen to be of
equal area while in a 3D model this should be equal volume.

The lowest resolution (10x40) contains 40 cells along the CMB, increasing across the

10 layers to 68 cells along the surface, for a global total of 539 cells (Fig. 2). The highest
resolution that we illustrate (20x160) then gives a global total of 4430 cells (Fig. 2). The
numerator $\rho_{c,j}$ for the proportion calculations (Eqs 2 & 3) for cells that do not contain a
particle ($\sum_{c=1}^{C} \rho_{c,j} = 0$) would cause a problem in its contribution to the entropy via the
natural logarithm. Note that $\lim_{x \to 0} x \ln x = 0$ and that therefore cells without a particle do not
add to any of the entropies, these cells are skipped in practice in the summation of Eqs (4-6).

## 3. Results


In this section, we describe the various obtained entropies. Starting with the particle

distribution $S_{pd}$. Next, we underline the importance of resolution for the local entropy $S_j$ in
our example model at different resolutions for the static composition distribution (case A)
and show how the local entropy evolves over time. Finally, we show the temporal evolution
of the global entropy $S$ for this model, which is also influenced by resolution and
compositional choices before we elaborate on the use of dynamic compositions (case B,
section 3.4).

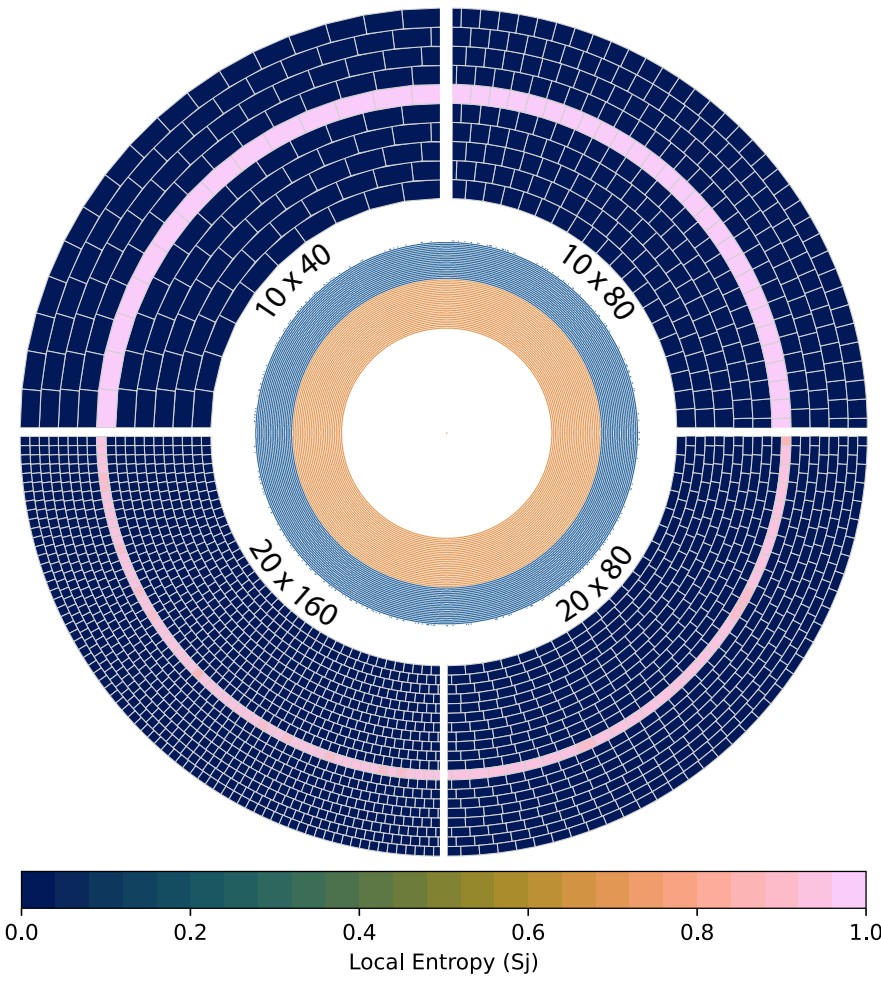


*Figure 2: Representation of the various tested resolutions. Shown is the initial (t=0 Ma) particle distribution (inner annulus)*
*and local entropy $S_j$ (outer annulus) for the static 50/50 composition distribution (case A). The cells with a high $S_j$ (pink)*
*indicate that both compositions are present, the unmixed cells (blue) contain only one composition.*
**3.1 Global particle distribution ($S_{pd}$)**

A total of ~96,000 of particles are initially distributed in a regular pattern (Fig. 2),

equally spaced throughout the annulus. Over time, these particles are passively advected
and their spatial distribution thus changes. The large number of particles in the initial
distribution provides a good coverage in all cells as quantified by the normalized global
entropy of particle distribution $S_{pd}$ which is at the modelling start close to 1 for both cases A
& B at the start (Fig. 3). As the initial composition ratios of case B are not equal (about 72%
lower mantle, 25% upper mantle, and 3% lithosphere) $S_{pd}$ is not 1 as for case A, but ~0.95,
still indicating that particles are distributed equally.
Over time, as particles are advected, the $S_{pd}$ does not change significantly for case A
in which particles cannot change composition, but it does change for case B (Fig. 3). This is
caused by the secular change in composition ratios in case B (Fig. 8 of van der Wiel et al.,
2023). $S_{pd}$ increases due to the increased percentage of lithosphere and upper mantle
particles in the domain. We tested the effect of cell resolution on $S_{pd}$ for both cases, which
does not show significant differences (Fig. 3). This indicates that the number of particles
used in our calculations is sufficient, also for our highest resolution (20x160).

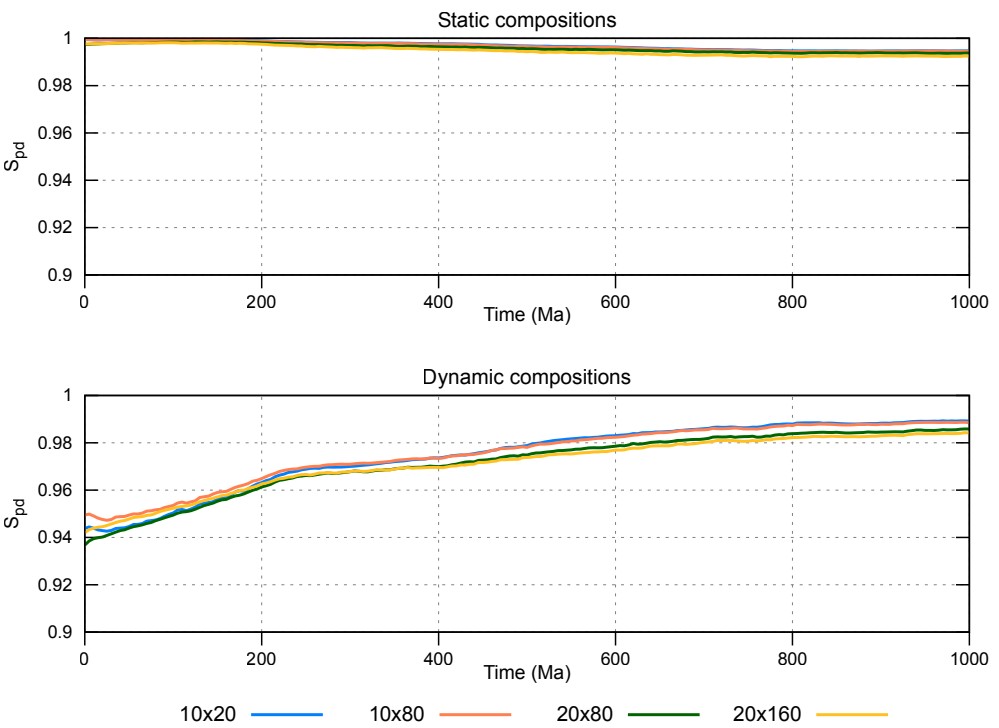


*Figure 3: Time evolution of $S_{pd}$ for the static (case A) and dynamic (case B) composition distributions of the four used*
*resolutions.*

**3.2 Local entropy ($S_j$)**

A local entropy of 1 means that the ratio of particle compositions within a cell is equal to the global composition ratio, e.g. in the initial distribution for case A (Fig. 2). $S_j = 0$ indicates that all particles in a cell have the same composition, although it does not indicate which composition. We illustrate the temporal evolution of particle distribution in 250 Ma steps (Fig. 4) for which we use the static particle composition ratio of case A and a cell resolution of 10x80 at the CMB (Fig. 3). After 250 Ma of convection evolution the initial distribution is undisturbed in most parts of the domain. The two compositions are only displaced since the onset of convection, but barely mixed. Mixing is concentrated around two major zones of downwelling where a narrow zone of single cells shows a local entropy $S_j$ that is non-zero (Fig. 4a).

At 500 Ma, some of the sharp boundaries between the two compositions have moved and a mixed boundary zone formed locally, reflected by the broader zone of non-zero local entropy (Fig. 4b). After 750 Ma, most of the upper mantle (top three cells) has $S_j > 0$ and zones in the lower mantle are mixed as well (Fig. 4c): the two starting compositions have been displaced and mixed through the mantle. At the end of the model, at 1000 Ma, the number of cells with non-zero $S_j$ in the upper mantle has decreased further, the zones of fully ($S_j \approx 1$) mixed lower mantle have increased in area. However, there are still zones of unmixed ($S_j = 0$) composition present. Unmixed initial 'lower' composition is preserved mainly in the mid-mantle while unmixed initial 'upper' composition is preserved near the CMB, i.e., this material sunk and was displaced, but did not mix (Fig. 4d).

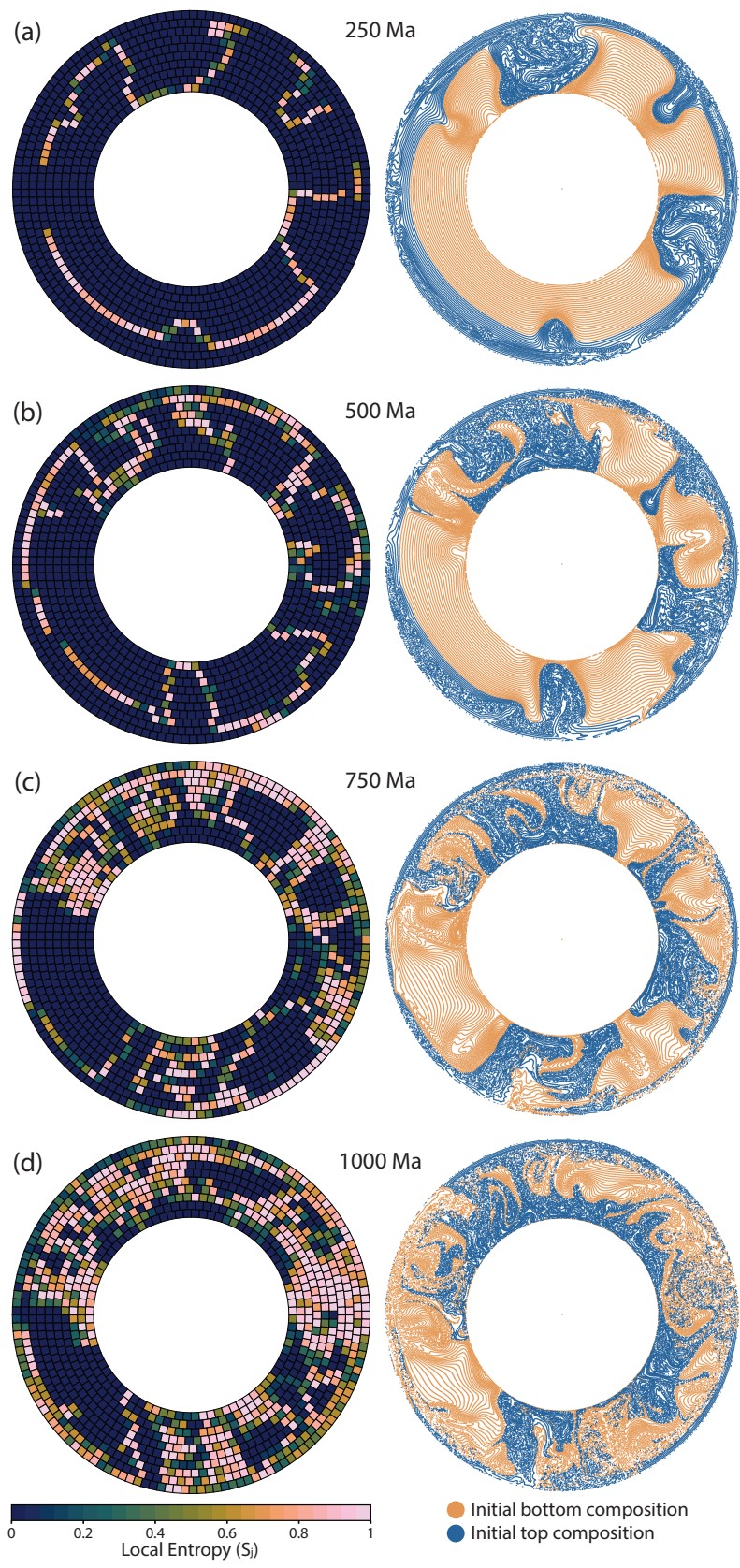


Figure 4 - Local Entropy $S_j$ (left) and particle distribution (right) at 250 Ma intervals of the model (model R - van der Wiel et al., 2023) with a static 50/50 ratio particle composition (case A) at a resolution of 10x80 cells.



Even though cell resolution does not significantly impact $S_{pd}$ it does affect the local

entropy $S_j$ (and thus also the global entropy, see next section). A smaller-sized cell mesh will
have fewer particles per cell, which increases the likelihood of sampling particles of only one
composition in zones with limited mixing, leading to zero local entropy. Doubling the angular
resolution from 10x40 to 10x80 shows on a global scale a similar trend after 1000 Ma of
convection: three zones of unmixed (low $S_j$) mantle separated by three zones of mixed
mantle (high $S_j$) (Fig. 5). However, it does show some increased detail in local entropy,
mainly in the 'mixed' zones of the model (Fig. 5). The large unmixed zones are of similar size
for these two resolutions, although the ratio of cells with a low $S_j$ compared to high $S_j$
changed. The larger unmixed zones are composed of initial 'lower' composition (Fig. 4d).

A radial increase in resolution, from 10 to 20 cells across the domain, refines the

calculation of local entropy. The number of cells with $S_j = 0$ becomes larger and increases
the size of the three main unmixed zones. At this resolution, $S_j$ resolves the 'continents':
thicker portions of lithosphere that were initially placed in the model (See Fig 2. of van der
Wiel et al., 2023). The 20x80 resolution has unmixed cells in regions that had high $S_j$ at
lower resolutions (Fig. 5). Finally, with the 20x160 mesh resolution, zones of initial upper and
lower composition (Fig. 4d) show up as low $S_j$ bounded by a single line of cells with high $S_j$
(Fig. 5). At this resolution the local entropy calculation resolves mantle structures such as the
boundaries between slabs and ambient mantle, showing mantle structure mapped into the
local entropy of mixing.

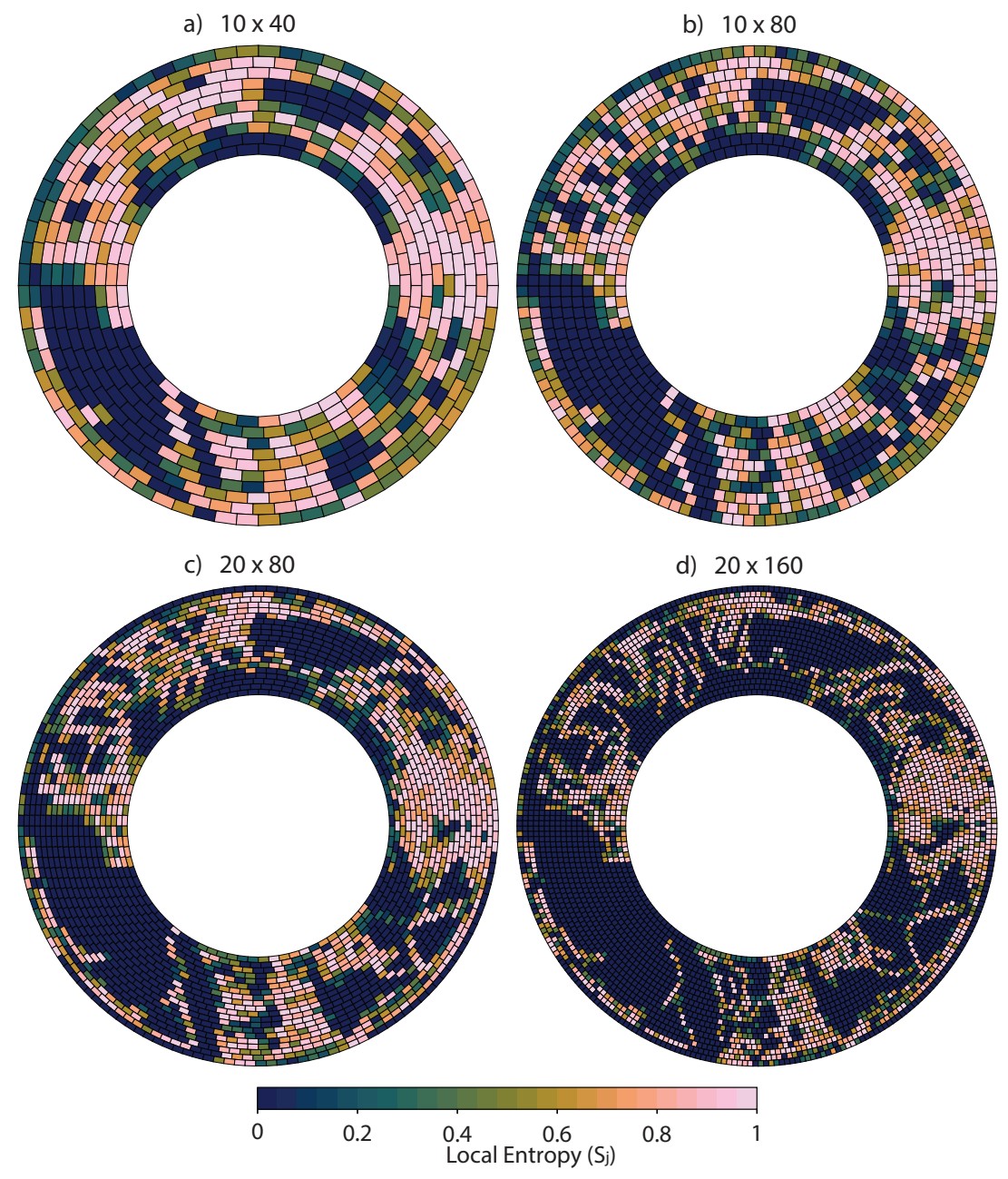

a)  10 x 40

b)  10 x 80

c)  20 x 80

d)  20 x 160

0     0.2     0.4     0.6     0.8     1
Local Entropy (Sj)


*Figure 5 - Local Entropy $S_j$ after 1000 Ma of mantle convection for the Reference model (van der Wiel et al., 2023) at four different resolutions of cells used to calculate the local entropy where b is identical as figure 4d.*

**3.3 Global entropy ($S$)**

The global entropy is a weighted average of the particle distribution proportion $P_j$ over cells and the compositional distribution within the cells $S_j$ (Eq. 6). Because the particle distribution irrespective of composition is almost equal to 1 in all tests (Fig. 2, section 3.1), we may consider the entropy $S$ as proxy for global compositional mixing. For the initial

distribution of composition based on depth, almost all cells have a local entropy $S_j = 0$,
apart from the cells that straddle the compositional boundary (Fig. 2). This distribution is an
unmixed state of the mantle and has a low global entropy, $S = 0.1$ for the resolutions with
10 radial levels and $S = 0.06$ for those with 20 radial levels (Fig. 6). The lateral resolution
does not matter for the initial distribution as the ratio of non-zero to zero $S_j$ cells is the
same.

While the mantle flow model evolves, compositions become more mixed and the

global entropy increases depending on mesh resolution, whereby smaller cells have a higher
chance to sample only one composition. Therefore, a higher resolution (smaller cells) yields
a lower global entropy after 1000 Ma of mantle convection: the 20x160 resolution yields $S =$
0.32 while the 10x40 resolution yields $S = 0.51$ (Fig. 6).

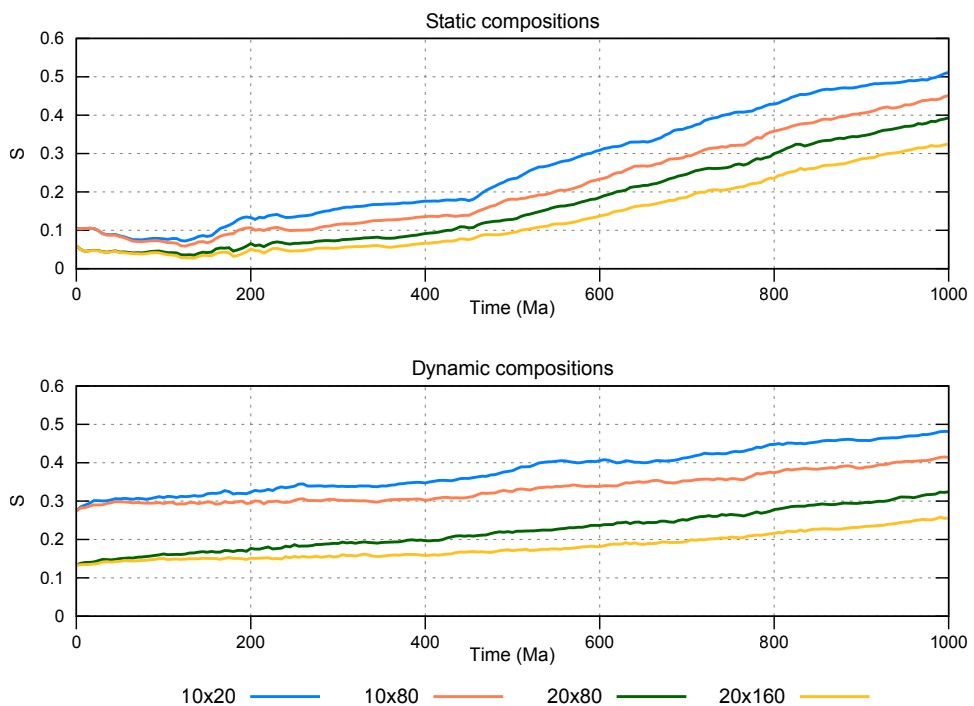


*Figure 6 - Global Entropy S of the model through time for different cell resolutions. Top: Case A (static compositions, bottom:*
*case B (dynamic compositions).*

**3.4 Case study: entropy with dynamic compositions**

Case B, which has dynamic compositions that depend on compositional evolution in the model, presents a practical application of the configurational Entropy. We track the entropy as the compositional ratios evolve and mix over time. The total number of particles that have been part of the lithosphere and subducted increases over time as new lithosphere and slab is being created while the volume of material that stays in the lower mantle decreases. In our example model, after 1000 Ma, the initial volumes of 3% lithosphere, 25% upper mantle, and 72% lower mantle have changed to 25% with lithosphere 'composition', 50% upper mantle and 25% lower mantle. In this example, the dynamic composition implies that no lower mantle composition exists above the 660 km-discontinuity and therefore the upper mantle cannot have a local entropy of 1. However, in parts of the lower mantle the three compositions are mixed where high local entropy is present. The parts of the domain containing subducted lithosphere are better mixed, indicative of the convective mixing behavior of our model. With the highest mesh resolution we can resolve the upper- to lower-mantle boundary in the local entropy as well as active and past locations of subduction (Fig. 7a).

The unmixed zones are of particular interest since they may provide direct information about compositions after 1000 Ma of convection. For all compositions there are cells with an unmixed signal, revealing the state of preservation of these compositions over time and over the whole domain. The entropy figures illustrate for instance the survival of unmixed original lower mantle material in the model, the fate of subducted lithosphere, and how upper mantle material is entrained downward during subduction (Fig. 7a).

This case has an entirely different local entropy than the static composition distribution of case A (Fig. 5). The dynamic case mainly focusses on the fate of subducted

lithosphere rather than global mixing of the upper and lower part of the domain. As in the
example with a static composition (case A), the global entropy $S$ for dynamic compositions is
also cell-size-dependent. The initial global entropy is higher than for static compositions as
there are now two compositional boundaries and over time the entropy only increases up to
$S = 0.25$ for the 20x160 resolution. For the 10x40 resolution, $S = 0.48$ after 1000 Ma which
is in the same range as the static two-composition example (Fig. 6).

Finally, we use the dynamic composition to illustrate how changing the vigor of

mantle convection changes the entropy. To this end, we compute the entropy after 1000 Ma
using a model in which much higher sinking rates of subducted slabs occurred than inferred
(model P - van der Wiel et al., 2023) and that consequently has faster mantle flow. Fig. 7b
illustrates that this model is much more mixed after 1000 Ma of convection than in model R
(Fig. 7a). It has cells with a local entropy close to 1 throughout much of the domain, unmixed
zones are smaller and located only in the top of lower mantle. Most of the local entropies
are in the mid-mixed range. This is because only 10% of the original 'lower mantle'
composition remains. Global entropy $S$ equals 0.42 for this model at the 20x160 resolution,
and even 0.60 for the 10x40 resolution, significantly higher than the reference model with
dynamic composition (Fig. 6). This example illustrates that the configurational entropy is
able to quantify mixing states in mantle convection models and is sensitive to overall
changes in model behavior.

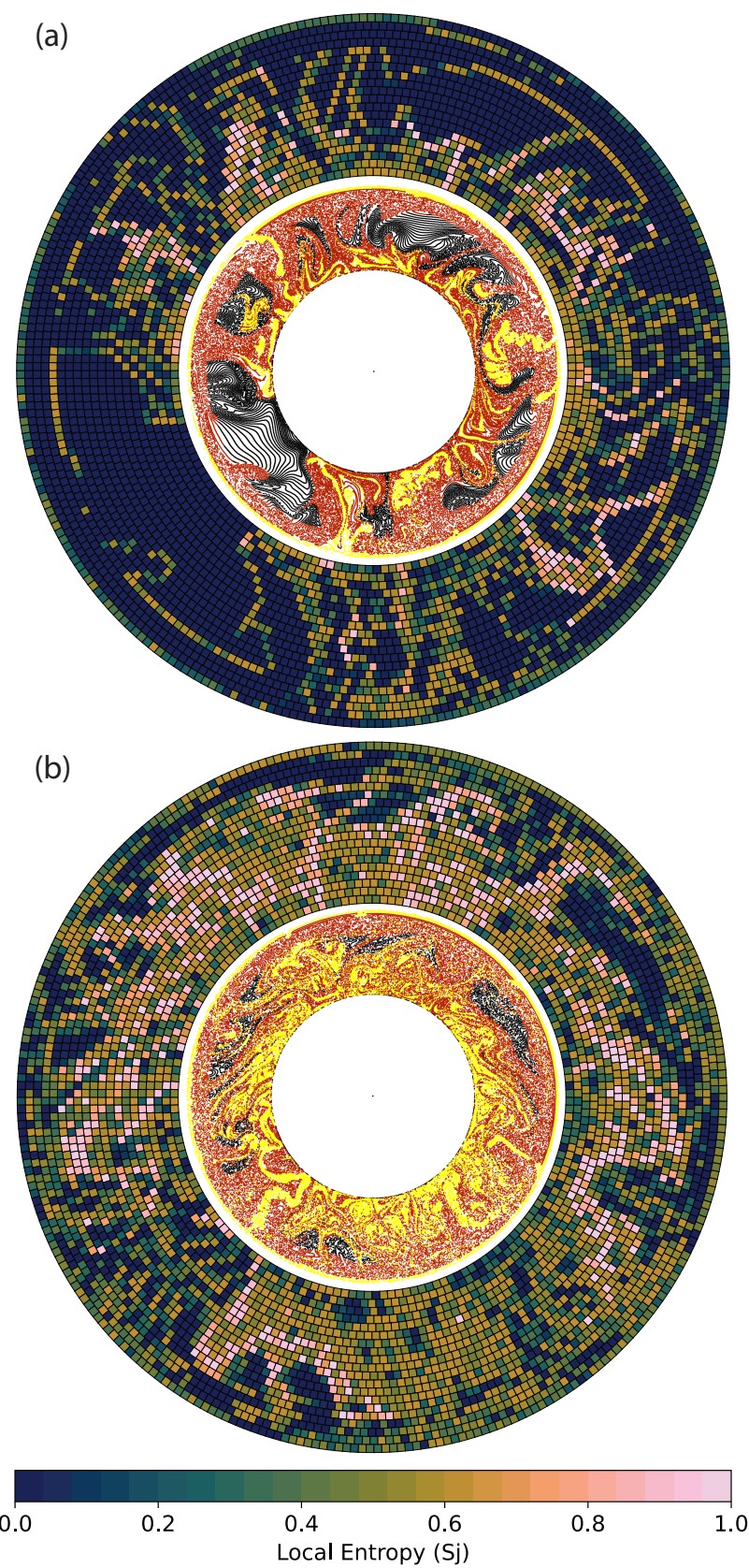

(a)

(b)

*Figure 7 - Local Entropy $S_j$ with a 20x160 resolution (outer annulus) and particle distribution (inner annulus) for dynamic compositions after 1000 Ma of simulated convection. Lower mantle (black), upper mantle (red) and lithosphere (yellow) compositions can change over time as function of depth. a) model R and b) model P with more vigorous convection of van der Wiel et al., (2023) as described in section 3.4.*

## 4. Outlook and conclusion

In this paper, we explore how configurational entropy may be applied to mantle convection models to quantify the degree of mechanical mixing, both on a local and global scale. Our results illustrate that entropy provides a way to track or map compositional heterogeneity over time using tracers or particles, which are commonly available in geodynamical models. Depending on the complexity of numerical models, any information that is stored on these tracers can be used to differentiate between 'compositions' used in the entropy calculations. The mantle convection models that we used to illustrate the use of configurational entropy were designed as numerical experiments to evaluate whether slab sinking rates scale with the vigor of mantle convection and mixing and did not aim to make a direct comparison between model and the real Earth. A direct comparison between configurational entropy and other measures used to quantify mixing, like the Lyapunov exponents (or time), is beyond the scope of this work. We see two arguments in favor of configurational entropy for specific uses: 1) its measurement does not require an integration over time thereby providing instant values for local and global entropy and 2) its flexibility, since the spatial distribution of any field carried by the particles, passive or active, such as chemical composition, water-content, reached depth or temperature, can be quantified.

For models that do make comparisons with the Earth, i.e. kinematically constrained by reconstructed plate motions and aiming to resemble Earth-like features (e.g., (Bull et al., 2014; Coltice & Shephard, 2018; Faccenna et al., 2013; Flament et al., 2022; Li et al., 2023; Lin et al., 2022) configurational entropy may serve as a means to quantify and map the degree of mixing of varying compositions, and hence to determine average cell composition, on a local, regional or a global scale. In such models the Lyapunov time would be useful to

quantify the deformation, or stretching, in the overall mantle (e.g., (Coltice, 2005; van Keken
& Zhong, 1999) or quantifying uncertainties in twin-experiments (e.g., (Bello et al., 2014;
Bocher et al., 2016).

On a global scale, such models (if run for 2-4 Ga) would for instance be able to track

volumes of material that have remained in the lower mantle during the evolution of Earth
(Fig. 7). These volumes are of interest, because they could explain the geochemical
detection of enigmatic primordial mantle, and feature in numerical models as the proposed
bridgmanite-enriched ancient mantle structures (BEAMS) of (Ballmer et al., 2017), or
surviving in the slab graveyard (Jones et al., 2021), or perhaps in LLSVPs or ULVZs
(Deschamps et al., 2012; Flament et al., 2022; McNamara, 2019; Vilella et al., 2021). In
addition, the use of entropy calculations may show how subducted lithosphere may become
stored in the mantle and to what degree original depleted and enriched crust, and slab
material mix with upper and lower mantle rock. Particularly, dynamically changing
compositions would benefit such studies, and in more sophisticated models that include
geochemical evolution (e.g., (Dannberg & Gassmöller, 2018; Gülcher et al., 2021),
geochemical reservoirs can be quantified with configurational entropy.

At smaller scales, entropy in mantle modeling is useful to track mixing at the scale of

a single subducting plate interacting with a mantle wedge, or a plume rising from the CMB.
This may be done based on location solely ($S_{pd}$), to track the dispersal of an initial cloud of
tracers in a slab or at the base of a plume (Naliboff & Kellogg, 2007), but also with the use of
composition through $S_j$ and $S$. For instance, it may quantify how different compositions of
material from the lowermost mantle are entrained by a plume and how material entrained
by that plume is mixed during its upward motion (e.g., (Dannberg & Gassmöller, 2018). For
instance, how material is mixed in the partially melting plume head, or in the partially
melting upper mantle below a ridge, mixing on the scale of a magma chamber may also be
mapped using configurational entropy, see (Perugini et al., 2015).

However, it may not yet be possible to numerically represent 3D mixing and motion

processes on all the scales illustrated above. In the end, the dynamics driving mantle
convection may force slow consumption and mixing away of primordial mantle by producing
lithosphere and plumes and mixing the geochemically segregated remains of these back into
the mantle. These processes lie at the basis for the widely recognized but still enigmatic
geochemical reservoirs that are thought to reside in the lower mantle such as those of
recycled continental crust (EM1, EM2), recycled oceanic crust (HIMU) (Yan et al., 2020),
recycled depleted lithospheric mantle (Stracke et al., 2019), and remaining primordial
mantle (Ballmer et al., 2017; Gülcher et al., 2020; Jackson et al., 2017). These processes also
culminate in the seismologically imaged mantle volumes of higher and lower seismic
velocity, or seismic attenuation, but the widely different scales at which geochemical and
seismological observations are made poses a problem to link such observations. Numerical
models may bridge these scales and eventually use our planets plate tectonic evolution to
predict the geochemical reservoirs as tapped by volcanoes, and mantle structure as imaged
by seismology. The configurational entropy in this paper may be helpful to quantitatively
determine where numerical models may successfully predict these seismological and
geochemical features.

## Appendix A

We here recall the equations of the manuscript and show the equations for normalization. Where

$n_{c,j}$ is the number of particles per composition $c$ in a cell $j$ and $N_c$ is the total number of c-

particles divided by the number of cells M. C represents the number of compositions used.

$$\rho_{c,j} = \frac{n_{c,j}}{N_c} \tag{1}$$

$$P_{j,c} = \frac{\rho_{c,j}}{\sum_{c=1}^{C} \rho_{c,j}} \tag{2}$$

$$P_j = \frac{\sum_{c=1}^{C} \rho_{c,j}}{\sum_{j=1}^{M} \sum_{c=1}^{C} \rho_{c,j}} \tag{3}$$

$$S_{pd} = -\sum_{j=1}^{M} P_j \ln P_j \tag{4}$$

$$S_j = -\sum_{c=1}^{C} P_{j,c} \ln P_{j,c} \tag{5}$$

$$S = \sum_{j=1}^{M} P_j S_j \tag{6}$$

$$S_{pd\ normalized} = \frac{S_{pd}}{\ln M} \tag{7}$$

$$S_{j\ normalized} = \frac{S_j}{\ln C} \tag{8}$$

$$S_{normalized} = \frac{S}{\ln C} \tag{9}$$

Four examples are given below, each with different distributions of particles and
compositions in a small rectangular grid of 4 cells. We use these four examples to illustrate
how the configurational entropy is affected by certain distributions. The background of the
cells is colored according to $S_j$ in grayscale, from 0 (black) to 1 (white) and the tracers shown
are randomly given a position in the cell appointed to them.

**Example 1 – equal distribution, fully mixed**

We start with a uniform distribution of particles with completely mixed compositions in each cell. The number of expected particles per composition per cell ($N_c$) is equal to the sum of the number of particles in that cell, this also reflected in vector $P_j$ which is equal for all cells – and therefore $S_{pd}$ is equal to 1 (after normalization) indicating a uniform distribution. The local entropy $S_j$ per cell is defined through $P_{j,c}$ which is equally distributed and equal to the normalization. Therefore, it indicates perfect mixing for all four cells. The global entropy combines $P_j$ and $S_j$ and is therefore equal to the endmember, which is 1.

$$n_{c,j} = \begin{pmatrix} 3 & 3 & 3 & 3 \\ 3 & 3 & 3 & 3 \end{pmatrix}$$

$$N_c = \begin{pmatrix} 3 \\ 3 \end{pmatrix}$$

$$\rho_{c,j} = \begin{pmatrix} 1 & 1 & 1 & 1 \\ 1 & 1 & 1 & 1 \end{pmatrix}$$

$$P_{j,c} = \begin{pmatrix} 1/2 & 1/2 & 1/2 & 1/2 \\ 1/2 & 1/2 & 1/2 & 1/2 \end{pmatrix}$$

$$P_j = \begin{pmatrix} 1/4 & 1/4 & 1/4 & 1/4 \end{pmatrix}$$

$$S_{pd} = 1.38629$$

$$S_j = \begin{pmatrix} 0.693 & 0.693 & 0.693 & 0.693 \end{pmatrix}$$

$$S = 0.69314$$

$$S_{pd\ normalized} = \frac{1.38629}{\ln 4} = 1$$

$$S_{j\ normalized} = \begin{pmatrix} 1 & 1 & 1 & 1 \end{pmatrix}$$

$$S_{normalized} = \frac{0.69314}{\ln 2} = 1$$

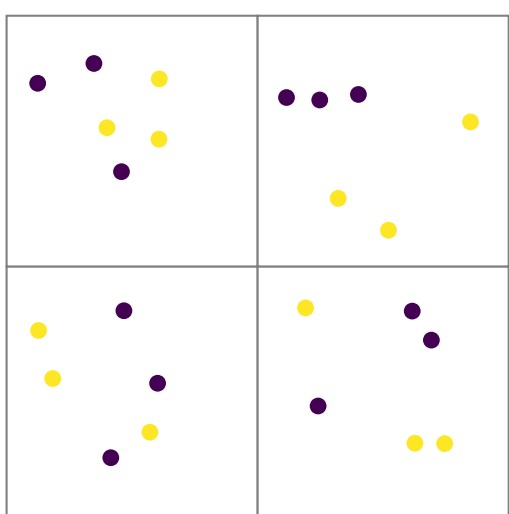

**Example 2 – equal distribution, no mixing**

The spatial distribution of particles is the same as in example 1 but compositions are not mixed, so $S_{pd}$ is still 1. $P_{j,c}$ is either one or zero per composition which both will lead to a zero for the local entropy which is therefore 0 for all four cells. As this local entropy feeds into the global entropy $S$, that is also 0.

$$n_{c,j} = \begin{pmatrix} 0 & 0 & 4 & 4 \\ 4 & 4 & 0 & 0 \end{pmatrix}$$

$$N_c = \begin{pmatrix} 2 \\ 2 \end{pmatrix}$$

$$\rho_{c,j} = \begin{pmatrix} 0 & 0 & 2 & 2 \\ 2 & 2 & 0 & 0 \end{pmatrix}$$

$$P_{j,c} = \begin{pmatrix} 0 & 0 & 1 & 1 \\ 1 & 1 & 0 & 0 \end{pmatrix}$$

$$P_j = \begin{pmatrix} 1/4 & 1/4 & 1/4 & 1/4 \end{pmatrix}$$

$$S_{pd} = 1.38629$$

$$S_j = \begin{pmatrix} 0 & 0 & 0 & 0 \end{pmatrix}$$

$$S = 0$$

$$S_{pd\ normalized} = \frac{1.38629}{\ln 4} = 1$$

$$S_{j\ normalized} = \begin{pmatrix} 0 & 0 & 0 & 0 \end{pmatrix}$$

$$S_{normalized} = \frac{0}{\ln 2} = 0$$

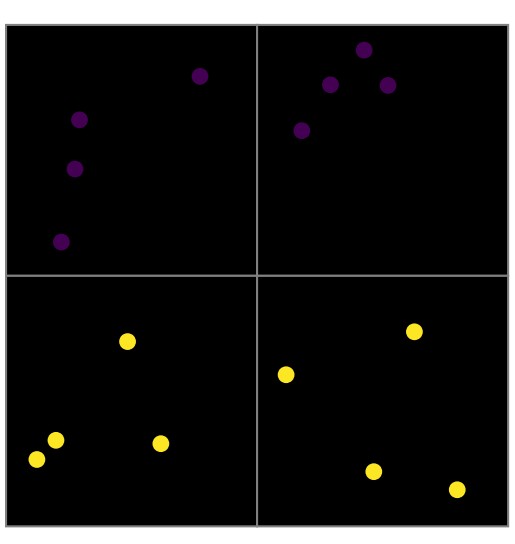

**Example 3 – random example with 3 compositions**

The distribution is ideally mixed as the expected number of particles in each cell is 3.5. $P_j$ is therefore not the same in each cell, but close to that. $S_{pd}$ is therefore close to 1 in this example. The compositions are not equally distributed, the top left cell is close to the expected distribution ($N_c$) and therefore has a local entropy close to 1 (after normalization by $ln(3)$). The bottom cells are equally far off expected values (1.5 off for purple, and 1 for the other colors) and have therefore the same $S_j$. As three cells have local entropy of about 0.5, but distributions are somewhat equal, the global normalized entropy is 0.678 – this reflects the weighted average of $S_j$ which is the global entropy.

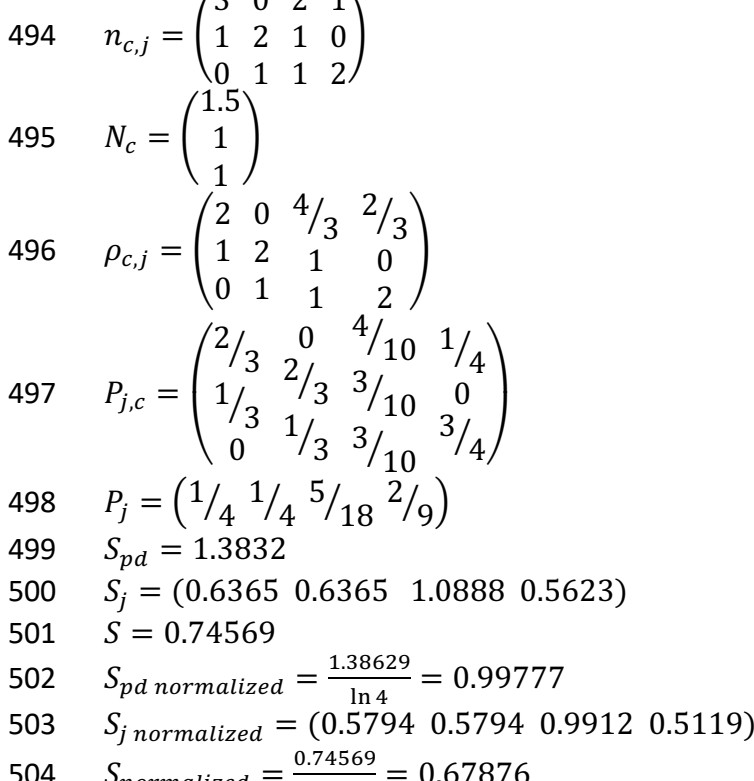

$$n_{c,j} = \begin{pmatrix} 3 & 0 & 2 & 1 \\ 1 & 2 & 1 & 0 \\ 0 & 1 & 1 & 2 \end{pmatrix}$$

$$N_c = \begin{pmatrix} 1.5 \\ 1 \\ 1 \end{pmatrix}$$

$$\rho_{c,j} = \begin{pmatrix} 2 & 0 & 4/3 & 2/3 \\ 1 & 2 & 1 & 0 \\ 0 & 1 & 1 & 2 \end{pmatrix}$$

$$P_{j,c} = \begin{pmatrix} 2/3 & 0 & 4/10 & 1/4 \\ 1/3 & 2/3 & 3/10 & 0 \\ 0 & 1/3 & 3/10 & 3/4 \end{pmatrix}$$

$$P_j = \begin{pmatrix} 1/4 & 1/4 & 5/18 & 2/9 \end{pmatrix}$$

$$S_{pd} = 1.3832$$

$$S_j = (0.6365 \quad 0.6365 \quad 1.0888 \quad 0.5623)$$

$$S = 0.74569$$

$$S_{pd\ normalized} = \frac{1.38629}{\ln 4} = 0.99777$$

$$S_{j\ normalized} = (0.5794 \quad 0.5794 \quad 0.9912 \quad 0.5119)$$

$$S_{normalized} = \frac{0.74569}{\ln 3} = 0.67876$$

**Example 4 – no equal distribution**

This last case showcases an uneven particle distribution, with an expected number of particles of 16.75 (sum $N_c$) that is not recovered in any cell. The vector $P_j$ is therefore not balanced and $S_{pd\ normalized}$ is 0.689, indicating an imperfect particle distribution. The top left is obviously unmixed with $S_j = 0$. The bottom cells have the same ratio of compositions and therefore a similar high $S_j$, as the 50/50 compositional ratio is not too far off the ideal ratio. The bottom cells contribute significantly to the global entropy $S$ and the top left cell has a sizable weighing factor ($P_3 = 0.238$) but as its $S_j = 0$ it does therefore not contribute to the total entropy.

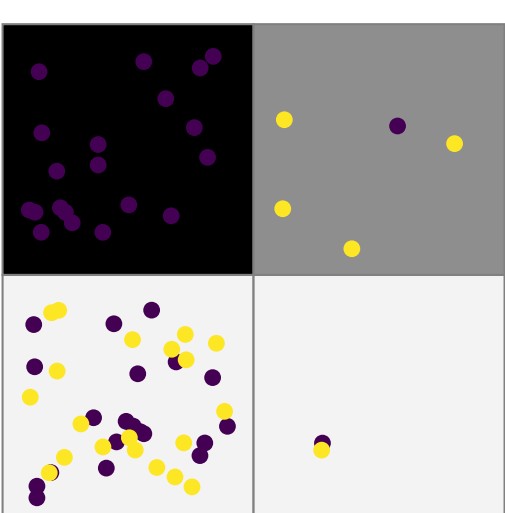

$$n_{c,j} = \begin{pmatrix} 20 & 1 & 20 & 1 \\ 20 & 1 & 0 & 4 \end{pmatrix}$$

$$N_c = \begin{pmatrix} 10.5 \\ 6.25 \end{pmatrix}$$

$$\rho_{c,j} = \begin{pmatrix} 1.9 & 0.095 & 1.905 & 0.095 \\ 3.2 & 0.16 & 0 & 0.64 \end{pmatrix}$$

$$P_{j,c} = \begin{pmatrix} 0.373 & 0.373 & 1 & 0.129 \\ 0.627 & 0.627 & 0 & 0.871 \end{pmatrix}$$

$P_j = (0.638\ 0.0319\ 0.2381\ 0.0919)$
$S_{pd} = 0.9576$
$S_j = (0.66\ 0.66\ 0\ 0.385)$
$S = 0.478$
$S_{pd\ normalized} = \frac{0.9576}{\ln 4} = 0.6907$
$S_{j\ normalized} = (0.953\ 0.953\ 0\ 0.556)$
$S_{normalized} = \frac{0.478}{\ln 2} = 0.689$

**Code availability**

The code that is used to create the appendix which calculates all the appropriate values can be found

online at:

https://github.com/cedrict/fieldstone/blob/master/python_codes/fieldstone_137/ministone.py

**Data availability**

The data used to make the figures are available on Zenodo (https://zenodo.org/records/10077983)

**CRediT authorship contribution statement**

**EvdW**: Conceptualization, Methodology, Investigation, Writing – Original draft, Visualization **CT**:

Methodology, Writing – Review & Editing **DJJvH**: Supervision, Writing – Review & Editing

**Competing interests**

The authors declare that they have no conflict of interest.

**Acknowledgements**

We thank W. Spakman for discussions. EvdW and DJJvH acknowledge the support by the Netherlands

Organisation for Scientific Research through NWO Vici Grant 865.17.001.

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
