# Peer review of "Quantifying mantle mixing through configurational Entropy"

_EGUsphere, 2023_

## Author Comment (AC1)

**General Comments**

This is a compact contribution that re-introduces the idea of configurational entropy to quantify mixing to the geosciences. The concept of configurational entropy to quantify mixing has been discussed in the geoscience literature before, and citations to such work are provided in the text. Where this extends on earlier work is that it extends the equations to cases where one has more than two types of objects / particles. This is a potentially very useful extension. They also go through some simple cases in the appendix which can help solidify the reader's understanding. The manuscript is concise and well written.

The work, after defining the equations for configurational entropy in such cases, applies it to 2D annulus mantle convection simulations. These simulations have already been published elsewhere. These examples show how the global entropy measure quantifies mixing in a single number – and shows visually that both the local and global measure of configurational entropy correspond qualitatively with the amount of mixing observed. The work is not sufficiently extensive to talk in detail about the implications for mantle mixing, and the authors do not attempt that, but it does allow them to speculate about future uses.

It is unclear how useful this measure will be though. We will need to wait to see how researchers use it. It is also unclear what the absolute value means, and it is dependent on spatial resolution (as mentioned by the authors).

We thank the reviewer for the comments provided and we are pleased that our approach and aim of the manuscript came across.

The contribution could have taken the opportunity discuss how the work here relates to the wider body of mantle mixing studies (e.g. Kellogg and Turcotte, McKenzie, Davies and Gurnis, Ferrachat and Ricard, Samuel and Farnetani, Tackley, van Keken, Olson, etc) . This literature is extensive, and some of it relates (indirectly) to configurational entropy. While these earlier studies do need to be better recognised here, careful consideration should be given as to the extent of additional description.  While significant additional description of earlier mixing studies would make for a more rounded contribution, especially if the discussion section tried to draw relationships from this work with earlier work introduced in an extended introduction. Such additional material though could potentially detract the reader from what is now a tight and focussed contribution. Therefore if the authors cannot see a way to make this a much more significant contribution by relating it to earlier work, then I would suggest that they restrict themselves to succinctly acknowledging and summarising the earlier work in this general field of mantle mixing to maintain its clear and concise form.

We agree with the reviewer that previous studies regarding mantle mixing have not been mentioned enough in our manuscript. We therefore added numerous examples of mixing studies to our introduction (see excerpt below) to provide the reader with a broader overview of quantifying mixing. However, except for the ones already cited in the paper, none of these additional studies use entropy as a measure and we feel comparing the various methods in detail goes beyond the scope of our study of reintroducing configurational entropy to the geoscience community.

*Excerpt 1: "While mixing technically involves diffusion at small scales and the term stirring has been proposed to account for the mechanical stretching and folding (Farnetani & Samuel, 2003), which is*

*infact our interest here, we shall nevertheless use the term mixing in the remainder of the manuscript as we use varying 'compositions' that are able to mix.*

*It has long been recognized that mantle convection is complex, and its mixing has been studied for decades, see (Kellogg, 1993; van Keken et al., 2003) for early reviews on this topic. Unsurprisingly, the advent of high-performance numerical modelling in the mid-90's saw a resurgence in the characterization of mantle mixing and its quantification. Various approaches have been proposed over the years, but the vast majority of these are based on the time evolution of a swarm of particles. Early studies (such as Hoffman & McKenzie, 1985; Olson et al., 1984a, 1984b; Richter et al., 1982; Schmalzl et al., 1996) use statistics to arrive at a mixing time scale. Another approach using the presence, addition, and/or removal of particles in a modelled domain is used to quantify mixing-times and degassing (sampling of primitive mantle) (Gottschaldt et al., 2006; Gurnis & Davies, 1986a, 1986b), to measure strain and the dispersal of tracers (Christensen, 1989; Kellogg & Turcotte, 1990) or to study the development of time-dependent mantle-heterogeneities (Hunt & Kellogg, 2001). Note that other methods have been proposed, such as a line method (Ten et al., 1998), a correlation dimension method (Stegman et al., 2002) and a hyperbolic persistence time method (Farnetani & Samuel, 2003).*

*More recently another approach has dominated the mantle mixing literature: it consists in measuring the Lyapunov time, which is the characteristic timescale for which a dynamical system is chaotic, or rather its inverse the Lyapunov exponent. It can be shown that mixing is laminar or turbulent by evaluating the Lyapunov exponent, the larger the exponent the more efficient the mixing is. A typical example uses a steady state velocity pattern obtained in a 3D spherical domain to advect passive particles (van Keken & Zhong, 1999). They use a very common approximation to the Lyapunov exponent, i.e., the Finite Time Lyapunov Exponent, which is based on the evaluation of the distance between a multitude of particle pairs that are initially very close to each other (i.e., stretching of this original distance after 4 Ga). This shows a strong diversity in mixing behavior dependent on the mantle flow characteristics. Other studies that used the same approach in studying a variety of mantle convection problems include: (Bello et al., 2014; Bocher et al., 2016; Colli et al., 2015; Coltice, 2005; Coltice & Schmalzl, 2006; Farnetani et al., 2002; Farnetani & Samuel, 2003; Ferrachat & Ricard, 1998, 2001; Samuel et al., 2011; Tackley & Xie, 2002; Thomas et al., 2024)."*

Overall, my assessment is that this is a useful contribution that deserves to be in the literature but only time will tell how significant it really will be. In this context we note that uses to date of earlier versions of configurational entropy have been limited in the geosciences, but maybe the additional flexibility of the measures presented here will encourage greater use.

**Specific Comments**
L 11 – Unclear how the measure can 'validate' a numerical model? Also unclear how can a model be 'validated' against local anomalies in the mantle inferred from other observations?
We agree with the comment and therefore changed the word 'validated' to 'compare'. We see an opportunity to test models that are driven by e.g., known plate motions to be compared with seismological or geochemical observations. For example, regional subduction models may cause the preservation of subducted lithosphere in the (lower) mantle. How such a slab interacts with the mantle, causing mixing in melts or in the seismic velocities of the mantle may be observed locally, and may also be tested/compared with an easily adaptable mixing quantity like configurational entropy.

L63/64 – While Shannon brought Entropy from a data perspective to people's attention, he did not talk about Configurational Entropy in that reference – nor how fast information on

compositional particle distribution is lost through flow. I accept that there is a relationship between standard configurational entropy and Shannon's information entropy. What is "fast information"? I think this sentence and reference needs a bit of work.

We have rephrased this introduction to entropy to make it clear:

*Excerpt 2: "Configurational entropy is analogous to the Shannon entropy (Shannon, 1948) and related to the probabilities derived from the distribution of particles with a certain value, i.e., composition. It can be used to track the mixing of particles independently of the physical process causing that mixing in numerical simulations as well as laboratory experiments."*

L79/80 – a bit strange to talk about – conditional probability – for a deterministic system. Maybe it is the conditional probability of finding this group of particles of composition c in cell j out of all other possible configurations. I appreciate that entropy related work is frequently discussed in terms of probability. Maybe it could instead be described as just something like the local proportion of particles of composition c (measured in terms of density) in cell j, relative to the total number of particles (measured in terms of density) in cell j.

See answer below.

L82 – again – maybe rather than the probability for the cell-sum – maybe a more deterministic description can be given here also. Is it just the proportion of all particles (again measured in terms of particle density) in cell j?

See answer below.

If this suggestion is taken up for describing these terms, I think it would also be OK to later or before include a statement pointing out that in statistical physics similar terms would be considered probabilities.

We understand that probabilities may indeed not have been the right choice of words for describing the calculated quantities, and we have taken the reviewer's advice by redefining them as proportions:

*Excerpt 3: "From the compositional density $\rho_{c,j}$ we calculate $P_{j,c}$ which is the proportion of particles of composition c in cell j relative to the total number of particles in the cell, both measured in terms of density through Eq. (2). We calculate $P_j$ through the cell-sum of all compositional densities in Eq. (3). $P_j$ is the proportion of the amount of particles in a cell relative to all the particles in the system. The quantities we describe here as proportions would be considered probabilities, or conditional probabilities, in statistical physics."*

L116/117 – The reference quoted states average RMS plate velocities over past 200 Ma of around 4 cm/yr, but your model R presents mean average surface velocities of around 2 cm/yr. I am not sure that is really close enough to say that it is in the range of reconstructed values. Maybe the sentence should be more specific – "the mean surface velocities in the model were x cm/yr, which can be compared with y cm/yr reconstructed in Zahirovic et al., 2015."

We have rephrased the sentence:

*Excerpt 4: "The surface velocity in the model were generally between 1 and 4 cm/a, which may be compared to the reconstructed values of 4 cm/a of Zahirovic et al., (2015)…"*

L288 – I feel that 'primordial' could be an emotive word here. For most whole Earth geoscientists, primordial suggests something that has survived since Earth's formation. I appreciate that 'primordial' here is taken to mean from the start of the simulation, but I think a more straightforward expression (with less 'baggage') could be used. Maybe 'original'. Speed readers might think that you have demonstrated that large regions of the lower mantle are likely to survive from Earth formation, not just 1000 Myr.

We agree that the term 'primordial' is too strong to be used in our model. We have changed it to 'original', as suggested, when discussing the results of our models. However, the occurrences of the word remain when discussing potential implications of the entropy on timescales larger than 1000 Ma and more Earth-like models, i.e., kinematically driven by reconstructed velocities.

**Technical Corrections**

We appreciate the thorough readthrough by the reviewer and all the suggestions given. We agree on all occasions and have changed the wording in the revised manuscript based on all the suggestions. Also including the definitions at the start of the appendix.

L 16 – 'stooled'?

L37 – 'entirely spatial' – missing word? Different?

L 48 – 'model the'– missing word? with?

L69 – not sure if this is a general definition of entropy. I think it is a definition of configurational entropy.

L75 'amount particles' -> amount of particles

L257 – 'spherical' resolution – unclear what you mean by spherical here? Do you mean lateral, or …?

L273 – "that has stays"?

L276-278 – As regards the "a local entropy of 1", it reads as if you mean the lower mantle composition - where? Anywhere? but that does not make sense - maybe you mean - " and therefore 'the local entropy above 660km' cannot have a local entropy of 1"?

check

L308-309 – 'illustrates ….. successfully quantifies mixing states'. While this might be suggested visually in a qualitative sense, I am not convinced that it has been shown in a quantitative way. I think it deserves a more accurate and weaker statement. Something that talks to the fact that this was a visual comparison that supports that configuration entropy gives the right ranking of mixing states.

Line 311. From what time in the simulation is this? I presume at the end?

Start of Appendix A – with the equations, I wonder whether $n_{c,j}$, $N_c$, $M$ and $C$ can be defined again here. Everything else is in the equations. It would then be complete.

Line 435 - "which is the global entropy is"

---

## Author Comment (AC2)

RC2:

The manuscript presents a geodynamic study that investigate the use of configurational entropy to characterise the mixing property of a model, and comparing a variety of models. One expressed goal is to use this measure to understand some geophysical and geochemical observations. Maybe it is because it is a paper about mixing, but I have a mixed analysis of the work. What I appreciate the most is the quality of the geodynamics models, of the data analysis and how the results are presented. It is impressive to have that in a paper, and it provides a lot of confidence in the outcomes. The shortcomings to me come from the target of the study, which is not yet found. My feeling is that it comes from the lack of inclusion of the literature about mantle mixing (maybe because most of it was published before 2010), and the duration of the simulation being 1 Gy that is too short to discuss primordial heterogeneities.

We appreciate the praise by the reviewer on how our manuscript is written. We agree that the manuscript is lacking mixing literature and accordingly added a summary of published mixing-studies using other quantifications in the introduction. We do not compare our results to geophysical or geochemical observations and we merely discuss how this entropy may be useful to compare model results with those observations. Therefore, we also changed the wording of 'primordial' to 'original' when presenting our results and only use it when discussing potential applications or other literature.

Excerpt 1: "While mixing technically involves diffusion at small scales and the term stirring has been proposed to account for the mechanical stretching and folding (Farnetani & Samuel, 2003), which is infact our interest here, we shall nevertheless use the term mixing in the remainder of the manuscript as we use varying 'compositions' that are able to mix.

It has long been recognized that mantle convection is complex, and its mixing has been studied for decades, see (Kellogg, 1993; van Keken et al., 2003) for early reviews on this topic. Unsurprisingly, the advent of high-performance numerical modelling in the mid-90's saw a resurgence in the characterization of mantle mixing and its quantification. Various approaches have been proposed over the years, but the vast majority of these are based on the time evolution of a swarm of particles. Early studies (such as (Hoffman & McKenzie, 1985; Olson et al., 1984a, 1984b; Richter et al., 1982; Schmalzl et al., 1996)) use statistics to arrive at a mixing time scale. Another approach using the presence, addition, and/or removal of particles in a modelled domain is used to quantify mixing-times and degassing (sampling of primitive mantle) (Gottschaldt et al., 2006; Gurnis & Davies, 1986a, 1986b), to measure strain and the dispersal of tracers (Christensen, 1989; Kellogg & Turcotte, 1990) or to study the development of time-dependent mantle-heterogeneities (Hunt & Kellogg, 2001). Note that other methods have been proposed, such as a line method (Ten et al., 1998), a correlation dimension method (Stegman et al., 2002) and a hyperbolic persistence time method (Farnetani & Samuel, 2003).

More recently another approach has dominated the mantle mixing literature: it consists in measuring the Lyapunov time, which is the characteristic timescale for which a dynamical system is chaotic, or rather its inverse the Lyapunov exponent. It can be shown that mixing is laminar or turbulent by evaluating the Lyapunov exponent, the larger the exponent the more efficient the mixing is. A typical example uses a steady state velocity pattern obtained in a 3D spherical domain to advect passive particles (van Keken & Zhong, 1999). They use a very common approximation to the Lyapunov exponent, i.e., the Finite Time Lyapunov Exponent, which is based on the evaluation of the distance between a multitude of particle pairs that are initially very close to each other (i.e., stretching of this original distance after 4 Ga). This shows a strong diversity in mixing behavior dependent on the mantle flow characteristics. Other studies that used the same approach in studying a variety of mantle convection problems include: (Bello et al., 2014; Bocher et al., 2016; Colli et al., 2015; Coltice,

*2005; Coltice & Schmalzl, 2006; Farnetani et al., 2002; Farnetani & Samuel, 2003; Ferrachat & Ricard, 1998, 2001; Samuel et al., 2011; Tackley & Xie, 2002; Thomas et al., 2024)."*

The first studies of mantle mixing date back to the 1980's and one of the difficulty was to deal with scales. So 2 approaches were taken: one dealing explicitly with scales, such as the authors do here with configurational entropy. It was in relationship with the observations of heterogeneities at all scales (marble cake like at first, more comprehensive nowadays with mid-ocean ridge heterogeneities etc...). The other approach is to work with the theory of dynamic systems, because mantle convection is chaotic. The use of Lyapunov exponent characterize the local and global properties of stretching of heterogeneities. Mixing happens when an heterogeneity is stretched at a scale at which chemical diffusion operates. This scale for the Earth is about 1cm I would say (some papers talk about it). Lyapunov exponent are a form of stretching rate when mixing is chaotic. I think Ferrachat and Ricard introduced it it 1998. Configurational entropy and Lyapunov exponent can be related with Ergodicity theory. It is fundamental to acknowledge the literature more extensively here and situate the innovation of the study. As the authors say, Naliboff and Kellogg in 2007 already used this measure, so the paper cannot really be the introduction of this measure. It has to be more than this (which I think it can be, but more explicitly). And it is also important to relate this measure to the Lyapunov exponent studies. At the end of the day, the stretching rate scales with the velocity in mantle convection. So faster velocities imply better mixing, which is what is observed here in the models as well.

We thank the reviewer for his comments regarding mixing studies from the 90's. Therefore, we have taken these suggestions above and the suggested authors at the end of the review and incorporated them in our manuscript. See the excerpt from the introduction in the manuscript above. We think that a full comparison between the Lyapunov exponent (or time) and the global entropy goes beyond the scope of this work as the global entropy also depends on the varying compositions in a model rather than just the mechanical stretching. We added a section to the discussion explain this point of view, see below. We must admit that we do not fully understand the Ergodicity theory and therefore refrain from mentioning this term. We have focussed the manuscript more on the additional variable of composition (which was not in Naliboff and Kellog 2007) rather than just stretching of the entire domain.

*Excerpt 2: "A direct comparison between configurational entropy and other measures used to quantify mixing, like the Lyapunov exponents (or time), is beyond the scope of this work. We see two arguments in favor of configurational entropy for specific uses: 1) its measurement does not require an integration over time thereby providing instant values for local and global entropy and 2) its flexibility, since the spatial distribution of any field carried by the particles, passive or active, such as chemical composition, water-content, reached depth or temperature, can be quantified."*

*Excerpt 3: "In such models the Lyapunov time would be useful to quantify the deformation, or stretching, in the mantle (e.g., van Keken & Zhong, 1999; Coltice 2005) or quantifying uncertainties in twin-experiments (e.g., Bello et al., 2014; Bocher et al., 2016)."*

There is an issue I think with the duration of the simulation. They are 1Gy long. It seems long, but it is not for a mixing study because (1) radiogenic geochemical heterogeneities needs time to develop and they mostly show preservation of >2Gy, (2) 1Gy is a little more than one mantle overturn today if we consider a slab takes 200 My to reach the bottom of

the mantle, (3) primordial mantle means it is almost as old as the Earth and (4) mixing/erasing heterogeneities at melting regions rates in the ancient Earth seemed much faster than today.

We agree that primordial is not right term to use in our model spanning 1 Ga, therefore we have changed it to 'original lower mantle material' when presenting our results. To use the term primordial one should indeed run models longer (e.g. 4 Ga) but also incorporate the mixing and evolution of mantle composition and the varying styles of convection and tectonics for example caused by higher temperatures than in today's mantle.

For the geodynamic community, 1Gy of preservation at the slow mantle convection rate of today is no surprise and does not necessarily teaches us about preservation of primordial heterogeneities. When working with Lyapunov exponent, it is important to run the simulation for a time sufficient to obtain finite time Lyapunov exponent. I am not sure 1 Gy would be sufficient here. So in this study, the setup and the methods are great, as good as they can be, but my appreciation is that the target of the study is not completely mature. I suggest refining the research objective, as aligning the study with previous work and integrating observations could significantly enhance its relevance.

We have focussed our contribution to stress the relevance of using varying compositions when using the entropy calculations, as it can also be used in regional models with mixing on shorter timescales, see excerpt 2.

Ultimately, this approach could elevate the study, making it a substantial contribution to the field. A minor detail: in cylindrical geometry the volume of the lower mantle is larger than in spherical geometry. To compare with Earth, maybe the geometry regions that mix in the study could be scaled by volume instead of depth? For the literature, I suggest to integrate papers by D. Turcotte, with L. Kellogg, also works of M. Gurnis and G. Davies, works of S. Ferrachat, Y. Ricard, of H. Samuel, of U. Christensen, P. Olson, P. Van Keken.

We have integrated the works of the suggested authors in the introduction and thank the reviewer once again for pointing those to us. We also added a remark regarding the cylindrical/spherical scaling in our set-up, for instance regarding equal volume cells when applied to 3D models.